behaviour/ecology/bioinformatics

supervised learning algorithms, accelerometer, sea turtle, animal-borne camera, behavioural classification, marine ecology

**Author for correspondence:**
Lorène Jeantet
e-mail: lorene.jeantet@iphc.cnrs.fr

# Behavioural inference from signal processing using animal-borne multi-sensor loggers: a novel solution to extend the knowledge of sea turtle ecology

Lorène Jeantet[1], Víctor Planas-Bielsa[2], Simon Benhamou[3], Sebastien Geiger[1], Jordan Martin[1], Flora Siegwalt[1], Pierre Lelong[1], Julie Gresser[4], Denis Etienne[4], Gaëlle Hiélard[5], Alexandre Arque[5], Sidney Regis[1], Nicolas Lecerf[1], Cédric Frouin[1], Abdelwahab Benhalilou[6], Céline Murgale[6], Thomas Maillet[6], Lucas Andreani[6], Guilhem Campistron[6], Hélène Delvaux[7], Christelle Guyon[7], Sandrine Richard[8], Fabien Lefebvre[1], Nathalie Aubert[1], Caroline Habold[1], Yvon le Maho[1,2] and Damien Chevallier[1]

[1]Institut Pluridisciplinaire Hubert Curien, CNRS–Unistra, 67087 Strasbourg, France
[2]Centre Scientifique de Monaco, Département de Biologie Polaire, 8 quai Antoine Ier, MC 98000 Monaco
[3]Centre d'Écologie Fonctionnelle et Évolutive, CNRS, Montpellier, France & Cogitamus Lab
[4]DEAL Martinique, Pointe de Jaham, BP 7212, 97274 Schoelcher Cedex, France
[5]Office de l'Eau Martinique, 7 Avenue Condorcet, BP 32, 97201 Fort-de-France, Martinique, France
[6]Association POEMM, 73 lot papayers, Anse a l'âne, 97229 Les Trois Ilets, Martinique
[7]DEAL Guyane, Rue Carlos Finley, CS 76003, 97306 Cayenne Cedex, France
[8]Centre National d'Etudes Spatiales, Centre Spatial Guyanais, BP 726, 97387 Kourou Cedex, Guyane

LJ, 0000-0001-7317-3154; VP-B, 0000-0003-0903-7603; FS, 0000-0002-5067-5896; CH, 0000-0002-6881-6546; DC, 0000-0002-2232-6787

The identification of sea turtle behaviours is a prerequisite to predicting the activities and time-budget of these animals in their natural habitat over the long term. However, this is

hampered by a lack of reliable methods that enable the detection and monitoring of certain key behaviours such as feeding. This study proposes a combined approach that automatically identifies the different behaviours of free-ranging sea turtles through the use of animal-borne multi-sensor recorders (accelerometer, gyroscope and time-depth recorder), validated by animal-borne video-recorder data. We show here that the combination of supervised learning algorithms and multi-signal analysis tools can provide accurate inferences of the behaviours expressed, including feeding and scratching behaviours that are of crucial ecological interest for sea turtles. Our procedure uses multi-sensor miniaturized loggers that can be deployed on free-ranging animals with minimal disturbance. It provides an easily adaptable and replicable approach for the long-term automatic identification of the different activities and determination of time-budgets in sea turtles. This approach should also be applicable to a broad range of other species and could significantly contribute to the conservation of endangered species by providing detailed knowledge of key animal activities such as feeding, travelling and resting.

## 1. Introduction

It is essential to assess the feeding behaviours of free-ranging animals in order to estimate their time budgets, and thus understand how these animals maximize their fitness [1,2]. However, investigating the foraging behaviour of sea turtles in their natural environment remains a significant challenge, as it is impossible to obtain long-term behavioural data through visual observations alone. Although some studies have provided relevant information on sea turtle diet through post-mortem stomach content analysis or the deployment of animal-borne video-recorders [3–5], the proportion of time that sea turtles allocate to feeding activities in the long term remains unknown. Time-depth recorders (TDR) have been used to record the dive profiles and durations of free-ranging sea turtles and have provided insights into their underwater activities [6–8]. However, a number of authors have underlined the limits of focusing on dive profile, as foraging activity cannot be distinguished from transit or resting phases [9,10]. The joint use of TDR and video-recorders revealed that the typical dive types described in [11,12] could not be associated with specific activities such as travelling, resting or foraging [13,14].

Devices combining miniaturized tri-axial accelerometers and TDR were described as a powerful tool to improve the identification of fine-scale behaviours in animals that cannot be easily monitored by visual observation [15–17]. Such devices have been deployed to study the behaviour and dive patterns of loggerheads (*Caretta caretta*, [18]), green turtles (*Chelonia mydas*, [19]) and leatherbacks (*Dermochelys coriacea*, [20]) during the inter-nesting period. However, the interpretation of the acceleration signals used in these studies to identify sea turtle behaviours in water was not validated by simultaneous visual observation, possibly resulting in misidentification and significant biases in the interpretation of the data.

A new approach was therefore necessary to reliably identify the underwater behaviours of free-ranging sea turtles without using direct visual observation (which is usually impossible) or video recordings, which are limited to short-term studies (a few hours) because of their high power consumption. Accelerometers permit the identification of feeding activity and time budget in marine animals such as seals and penguins by recording head movements that are likely to correspond to prey captures [21–23]. For the same purpose, accelerometers have been placed on the beak [24–27] or the top of the head [28] of sea turtles to record beak-openings and capture attempts. However, placing the device in this way is likely to result in a significant disturbance for the individuals and cannot be considered for long-term use (up to several weeks). It was therefore crucial to develop a protocol for the long-term recording and identification of sea turtle feeding activities that minimizes disturbance to the animals while making optimal use of the subtle variations in data acquired by loggers that are mounted on the carapace rather than the head.

Further work is needed to validate the identification of sea turtle underwater behaviours by data acquired by animal-borne sensors. In particular, before attempting to provide new insights about the at-sea behaviours of sea turtles in natural conditions, one needs to automatically and correctly identify these behaviours, including those that are hard to detect but play a key role such as feeding, from data acquired in a way that minimizes the disturbance of the equipped animal. The aim of our study is therefore to develop a new approach fulfilling this need. In this framework, we will use the results we obtained about turtles' behaviours only to illustrate the output of our approach without attempting to give them any biological significance. Although sea turtle behaviours have mainly been inferred from combined acceleration and depth data, the additional use of a gyroscope (which records angular velocity)

can provide further relevant information in remote behavioural identification [29–31]. Thus, we deployed loggers combining an accelerometer, a gyroscope and a TDR on the carapace of free-ranging immature green turtles. This equipment was linked to a video-recorder that was mounted in the logger device to provide visual evidence that could validate logger interpretations of behaviours, given that our approach ultimately aims to infer behaviours solely through logger use. Surface behaviours were identified separately from depth data. The study tested a set of methods to infer diving behaviours from the signals provided by the accelerometer, gyroscope and TDR, including automatic segmentation and supervised learning algorithms. The validity of our approach was tested through the use of confusion matrices and by comparing the inferred activity budgets with those obtained from video recordings.

# 2. Material and methods

## 2.1. Data collection from free-ranging green turtles

The fieldwork was carried out from February 2018 to May 2019 in Grande Anse d'Arlet (14°50′ N, 61°09′ W), Martinique, France. We deployed CATS (Customized Animal Tracking Solutions, Germany) devices for periods ranging from several hours to several days on free-ranging immature green turtles. A CATS device comprises a video-recorder (1920 × 1080 pixels at 30 frames s$^{-1}$) combined with a tri-axial accelerometer, a tri-axial gyroscope and a TDR (electronic supplementary material, figure S1). The maximum battery capacity was considered to provide a recording capacity of 18 h of video footage and 48 h for other data. These devices were programmed to record acceleration and angular velocity (gyroscope) at a frequency of 20 or 50 Hz according to the recording capacity of the logger (the 50 Hz data were subsampled at 20 Hz using a linear interpolation to homogenize the sample). Depth was recorded at 1 Hz using a pressure sensor with a range from 0 to 2000 m and 0.2 m accuracy.

The relatively shallow depths of the area allowed free divers to capture the turtles manually, as described in Nivière et al. [32]. Once an individual had been caught, it was placed on a boat and identified by scanning its passive integrated transponder (PIT) or tagged with a new PIT if it was unknown. It was then weighed and its carapace length was measured (electronic supplementary material, table S1). The device was attached to the carapace using four suction cups. Air was manually expelled from the cups, which were held in place by the use of a galvanic timed-release system. The dissolving of these releases by seawater and the positive buoyancy of the device (23.3 × 13.5 × 4 cm for 0.785 kg) led to the remote release of the device several hours later. Devices were recovered by geolocation of an Argos SPOT-363A tag (MK10, Wildlife Computers Redmond, WA, USA), which was glued to the CATS device, with a goniometer (RXG-134, CLS, France). Instruments were deployed on 37 individuals, but complete datasets including video, acceleration, gyroscope and depth values were only recovered for 13 individuals (electronic supplementary material, table S1).

## 2.2. Processing of video data and behavioural labelling

The video footage was watched to identify the various behaviours and determine their starting and ending times to the closest 0.1 s. Acceleration, angular velocity and depth data corresponding to each behavioural phase were visualized using R software (version 3.5.3) and the package *rblt* (figures 1 and 2; [33]). The 46 resulting behaviours were clustered into categories according to their similarities (the definition of the various behaviours is available in electronic supplementary material, table S2). We retained seven main expressed categories for the multi-sensor signals, namely 'Breathing', 'Feeding', 'Gliding', 'Swimming', 'Resting', 'Scratching' and 'Staying at the surface'. All other behaviours were very infrequent and were grouped in an eighth category labelled 'Other'.

## 2.3. Analysis of the angular velocity and acceleration data

The device was installed on the carapace in a tilted position along a longitudinal axis to obtain video images of the head. This results in biased values of accelerations and angular speeds for the surge (i.e. back-to-front) and heave (bottom-to-top) body axes, which therefore had to be corrected (see R-script in electronic supplementary material). The static acceleration vector (i.e. the component due to gravity) $\bar{\mathbf{a}} = (\bar{a}_x, \bar{a}_y, \bar{a}_z)$ was obtained by separately averaging the acceleration values ($a_x$, $a_y$ and $a_z$) on the surge, sway (right-to-left) and heave axes, respectively, over a centred running temporal window set to $\Delta t = 2$ s, which was the smallest window resulting in a norm $\|\bar{\mathbf{a}}\|$ that remains close to

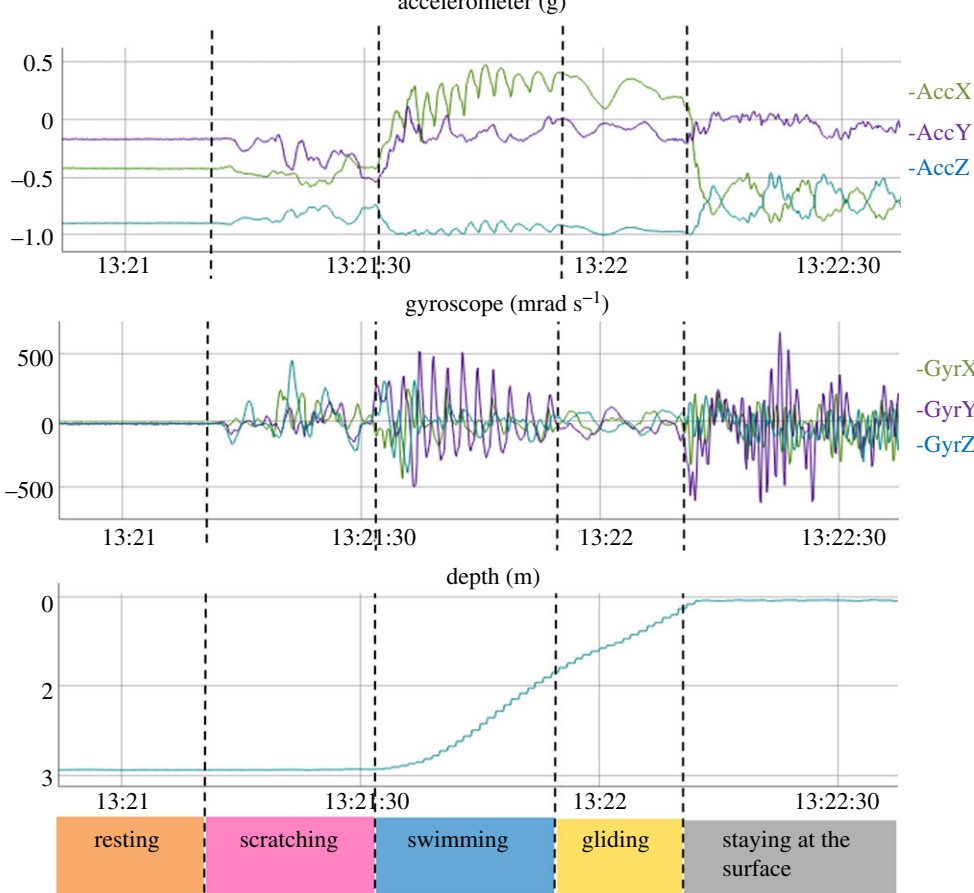

**Figure 1.** Raw acceleration, gyroscope and depth profiles for several behaviours expressed by turtle #12.

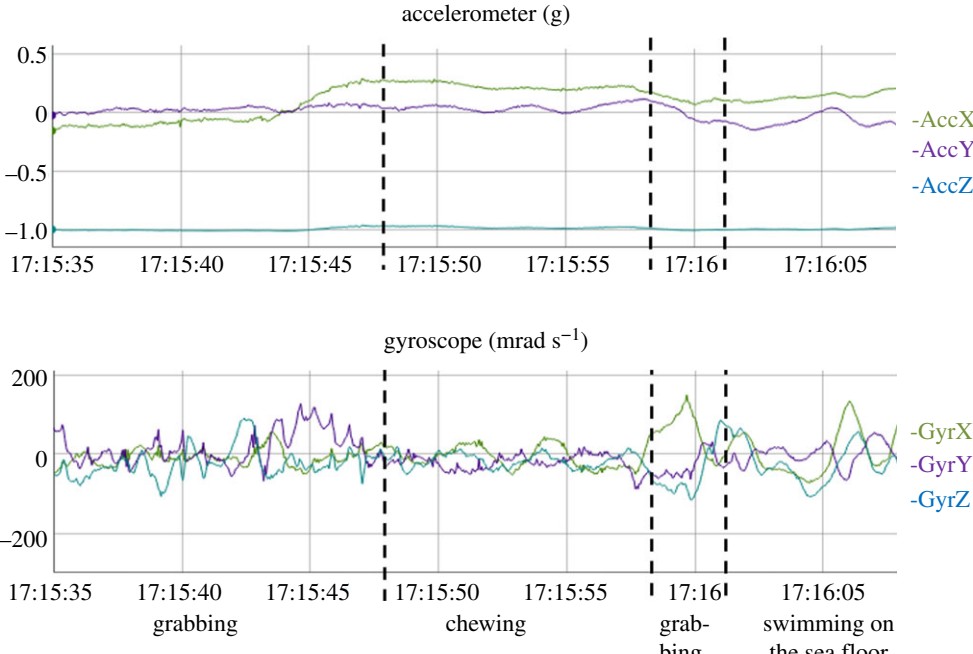

**Figure 2.** Raw acceleration and gyroscope signals obtained for the feeding behaviours expressed by turtle #6. The definitions of the behaviours are available in electronic supplementary material, table S2).

$1g$ (9.98 m s$^{-2}$) for almost all measures. The dynamic body acceleration was then computed as DBA = $\sqrt{\mathbf{d}^2}$, where $\mathbf{d} = \mathbf{a} - \bar{\mathbf{a}}$ is the dynamic acceleration vector [34]. Similarly, the rotational activity was computed as RA = $\sqrt{\mathbf{g}^2}$, where $\mathbf{g} = (g_x, g_y, g_z)$ is the angular velocity ($g_x$, $g_y$, and $g_z$ correspond, respectively, to the values of roll, pitch and yaw per unit time provided by the gyroscope).

## 2.4. Segmentation of the multi-sensor dataset

The automatic identification of the labelled behaviours from the multi-sensor signals required the segmentation of the dataset into homogeneous behavioural bouts. We started by relying on the depth data to distinguish the dives, defined as depths exceeding 0.3 m for at least 5 s, from the surface periods. We attributed the surface periods to either 'Breathing' or 'Staying at the surface', according to whether the turtle remained at the surface for less or more than 6 s, respectively. We then distinguished between the various possible diving behaviours by using a changepoint algorithm, the pruned exact linear time (PELT) algorithm (R package *changepoint*; [35]), in which the 'pen.value' parameter, which corresponds to the additional penalty in the cost function for each additional partition of the data, can be manually adjusted. We tested different values and retained those which resulted in the best balance between obtaining homogeneous behavioural bouts and limiting over-segmentation. We first detected depth changes over 3 s of each dive (function *cpt.mean*, penalty = 'Manual', pen.value = 5) to obtain segments which were labelled as 'ascending', 'descending' or 'flat' depending on whether the vertical speed was greater than 0.1 m s$^{-1}$, less than –0.1 m s$^{-1}$ or between these two values, respectively. These ascending and descending segments were further segmented based on the DBA mean and variance (function *cpt.meanvar*, penalty = 'Manual', pen.value = 50) in order to distinguish between the swimming and gliding phases of these segments. The green turtle is a grazing herbivore which mainly feed on seagrass and algae [36]. The head movements occurring during feeding activities are easily detected by gyroscopes and/or accelerometers set directly on the head, but are rarely detected when these sensors are placed on the carapace. We did, however, note that the carapace tended to display pitch oscillations when the turtle pulled on the seagrass, an activity that we refer to hereafter as 'Grabbing' (figure 2). Accordingly, we further segmented the 'flat' segments based on the variance of $g_y$ (angular speed in the animal's sagittal plane; function *cpt.var*, penalty = 'Manual', pen.value = 20) to pinpoint this behaviour. Each segment was then labelled as either the behavioural category that was expressed for at least 3/5 of its duration, or as 'Transition' if several behaviours were involved with none of them occurring for 3/5 of the behavioural bout. Thus, the overall procedure classified multi-sensor signals into nine categories comprised surface behaviours (Breathing and Staying at the surface) which were identified using depth data alone, diving behaviours (Feeding, Gliding, Resting, Scratching and Swimming) and also 'Other' and 'Transition', for which supervised learning algorithms were required.

## 2.5. Identification of the diving behaviours by supervised learning algorithms

We trained five supervised machine learning algorithms—(i) classification and regression trees (CART), (ii) random forest (RF), (iii) extreme gradient boosting (EGB), (iv) support vector machine (SVM), and (v) linear discriminant analysis (LDA)—to associate the seven diving behaviour categories with the corresponding patterns of different input variables. They are the most commonly used classifiers in behaviour recognition and are considered to be relevant in ecology studies [17,37,38]. These algorithms were applied to our data using the R packages *rpart* [39] for CART, *randomForest* [40] for RF ($n = 300$, mtry = 14), *xgboost* [41] for EGB (num_class = 7, eta = 0.3, max_depth = 3), *e1071* [42] for SVM and *MASS* [43] for LDA.

For each segment, the algorithms were fed with four descriptive statistics (mean, minimum, maximum and variance) computed for the three linear acceleration values ($a_x$, $a_y$ and $a_z$), for the three angular speeds values ($g_x$, $g_y$ and $g_z$), and for DBA and RA. We also included the difference between the last and first depth values, and the duration of each segment. The fact that 'Feeding' was characterized by high-frequency oscillations, in particular in terms of pitch speed (figure 2), but also (although less obviously) in terms of roll speed and surge/sway accelerations, enabled us to distinguish this behaviour from the others. We characterized these oscillations as follows: (i) we smoothed the high-frequencies values of $g_x$, $g_y$, $a_x$ and $a_y$ (after correction for the inclination) using a centred running mean over a 1 s window; (ii) for each of these four variables, we computed the differences $d$ between the raw values and their respective running means; (iii) we computed the mean value of these differences for the whole segment $m(d)$; (iv) we computed the mean and the maximum value of the squared differences $(d-m(d))^2$ and we added these two parameters to the list of variables used to feed the algorithms, i.e. 42 variables for each segment. Such a number of variables may be characterized by numerous correlations. However, machine learning algorithms are less sensitive than

classical regression methods to correlation in the explanatory variables. Nevertheless, for a simpler interpretation purpose, we looked for some possible reduced set of variables that may reach the same accuracy as the full dataset, but we did not find any convincing one. As the focus was more on predictability than interpretability (as is usual the case in machine learning), we kept all the 42 variables.

## 2.6. Validation of the automatic behavioural inferences

To estimate the ability of our procedure to correctly infer the behaviours of sea turtles based on acceleration, angular velocity and depth data, we repeatedly performed $2/3 : 1/3$ splits of the sample of 13 individuals, with nine individuals retained for the learning phase and the remaining four individuals used to validate the outcome. From the 715 possible combinations, we retained the 376 combinations in which 'Feeding' and 'Scratching' were not under-represented in the training dataset (i.e. when more than 60% of total feeding and scratching segments were present, i.e. 898 and 714, respectively). Nevertheless, the number of 'Feeding' and 'Scratching' segments was much lower than those attributed to 'Resting' and 'Swimming' (11 168 and 10 293 segments, respectively). As an unbalanced training dataset can hinder the performance of supervised learning algorithms [44], we set an upper limit at 1000 segments per behaviour for the training dataset. These segments were randomly selected for the over-expressed categories at each training trial.

For each trial, we evaluated the efficiency of the different methods by computing the number of well-identified behaviours (true positive, TP, and true negative, TN) and of behaviours considered to be misclassified (false negative, FN, and false positive, FP) into a confusion matrix. We calculated three indicators for each behaviour: (i) 'Sensitivity' = TP/(TP + FN), also called *true positive rate*, *hit rate* or *recall*, measures the ability of a method to detect the target behaviour among other behaviours; (ii) 'Precision' = TP/(TP + FP), also called *positive predictive value*, measures the ability of a method to correctly identify the target behaviour; and (iii) 'Specificity' = TN/(TN + FP), also called *selectivity* or *true negative rate*, measures the ability of a method to avoid wrongly considering other behaviours as the target behaviour. We also computed 'Accuracy' = (TP + TN)/(TP + TN + FP + FN), which measures the ability of a method to correctly identify all behaviours as a whole.

Furthermore, to possibly improve the performance and/or minimize the variance of behavioural inferences, we also relied on the 'Ensemble Methods' [45,46], which consisted of combining the results obtained with the five supervised machine learning algorithms. We tested two such methods. The first was the 'Voting Ensemble' (VE), which retained the most frequently predicted behaviour. The second involved a 'Weighted Sum' (WS), where weights were given to the different predicted behaviours, based on 'Precision' (weighting based on Sensitivity and Specificity was also tested but gave poor results). In order to highlight the best method to automatically identify the diving behaviours and particularly the feeding behaviours, we used ANOVA to compare the mean global accuracy obtained for the 376 combinations of the seven classifiers (CART, SVM, LDA, RF, EGB, VE and WS). As the result of the ANOVA showed significant effects, we ran pairwise comparisons of mean performance using the Tukey HSD test.

Finally, the individual activity budgets were inferred by computing the proportion of time involved in the various surface behaviours (Breathing and Staying at the surface) inferred from depth data, and the proportion of time dedicated to diving behaviours (Feeding, Gliding, Other, Resting, Scratching and Swimming), inferred using the best classifier (figure 3). The inferred activity budgets were compared to those obtained with video recordings.

## 3. Results

A total of 66.2 h of video were recorded, with a maximum of 14.6 h for one individual (table 1). The seven specific behavioural categories retained for the analysis (Breathing, Feeding, Gliding, Resting, Scratching, Staying at the surface and Swimming) represented 99% of the total duration. Only the two shortest deployments were not associated with a feeding event while the maximum duration of feeding represented only 8% of the recording time of the individual. The catching of jellyfish was observed only occasionally in three individuals. This behaviour represented only 0.1% of the total feeding duration of the 13 individuals, whilst the rest of the feeding time for those individuals was used for grazing on seagrass. For the others, feeding consisted only of grazing on seagrass. 'Scratching' was particularly expressed by one turtle, and represented 13% of its observation time.

The seven classifiers identified the five specific behavioural categories on which we focused (Feeding, Gliding, Resting, Scratching and Swimming) and two additional categories, 'Transition' and 'Other', with an accuracy ranging from 0.91 to 0.95. The highest score was obtained with WS and the lowest one with

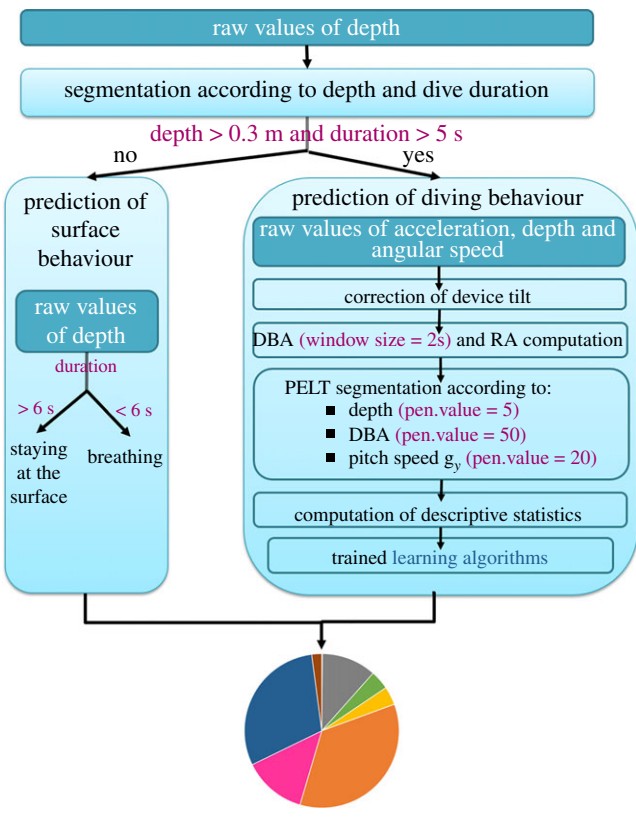

**Figure 3.** Workflow of automatic behavioural identification using acceleration, angular speed and depth data, as adapted to the green turtle. The hyper-parameters set-up specifically for green turtle data are highlighted in pink. The application of this workflow for other marine species would necessitate the identification of the optimal hyper-parameter values for each species.

SVM. The Tukey HSD test indicated that the RF, VE and EGB outputs were not significantly different (0.935, 0.932 and 0.932, respectively). All classifiers identified the behavioural category with a low false positive rate (less than 0.1 for the best classifiers; figure 4). Few segments were wrongly identified as 'Feeding' with the WS method, which thus obtained the lowest false positive rate (with respect to other classifiers) for this behaviour. The best true positive rates, for the seven classifiers, were obtained in the 'Scratching' category despite its low occurrence in the dataset, meaning that this behaviour was relatively well identified when it occurred.

The activity budget, representing the percentage of the total mean time allocated to each behavioural category, showed similar proportions between the predictions and the observations (figures 5 and 6). This result highlights the ability of our method and the WS model to predict the behaviours of immature green turtle in natural conditions. The main differences between the observed and predicted activity budgets were seen in the 'Resting' and 'Swimming' behaviours (figure 5 and 6). These differences were small and represented less than 3% of the total observed time (table 2). 'Feeding' and 'Scratching' were under-represented in our models and consequently their difference between predicted–actual time represent roughly 1% of the total observed time. Their low expression for some individuals led to an important percentage difference with respect to the observed time of the behaviour even if they were predicted in small proportion. The results obtained for each individual are available in electronic supplementary material, table S3. With a very low true positive rate, the predicted time of 'Transition' represented on average 0.2% of the total observation time. Thus the overall procedure was able to reliably infer the seven mainly expressed behaviours of the immature green turtles.

# 4. Discussion

This is the first study to validate the use of acceleration, gyroscope and TDR signals for inferring free-ranging green turtle behaviours. In previous studies, carapace-mounted accelerometers were used to

**Table 1.** Total duration (seconds) of the observed sequences of behavioural categories for the 13 free-ranging immature green turtles.

| behaviour | #1 | #2 | #3 | #4 | #5 | #6 | #7 | #8 | #9 | #10 | #11 | #12 | #13 | total | relative importance (%) |
|---|---|---|---|---|---|---|---|---|---|---|---|---|---|---|---|
| breathing | 36 | 301 | 37 | 20 | 66 | 87 | 89 | 6 | 57 | 27 | 132 | 75 | 293 | 1226 | 0.51 |
| feeding | — | 1499 | 162 | 540 | 152 | 1955 | 70 | — | 1030 | 661 | 6 | 28 | 178 | 6281 | 2.64 |
| gliding | — | 896 | 366 | 524 | 211 | 284 | 1054 | 102 | 1257 | 129 | 609 | 372 | 2271 | 8075 | 3.39 |
| resting | — | 10 134 | 7747 | 4760 | 5807 | 11 302 | 19 502 | 711 | 7190 | 602 | 17 579 | 3814 | 27 441 | 116 590 | 48.95 |
| scratching | — | 512 | 574 | 1789 | 8 | 903 | 136 | — | 64 | 21 | 177 | 94 | 218 | 4496 | 1.89 |
| staying at the surface | — | 898 | 1396 | 1546 | 1394 | 2541 | 2955 | 573 | 1032 | 818 | 1485 | 582 | 3246 | 18 465 | 7.75 |
| swimming | 5279 | 6522 | 3801 | 4082 | 6421 | 6005 | 4800 | 2026 | 6354 | 5493 | 7739 | 2760 | 18 895 | 80 178 | 33.66 |
| other | — | 258 | 169 | 283 | 148 | 818 | 136 | 45 | 261 | 209 | 233 | 140 | 188 | 2887 | 1.21 |

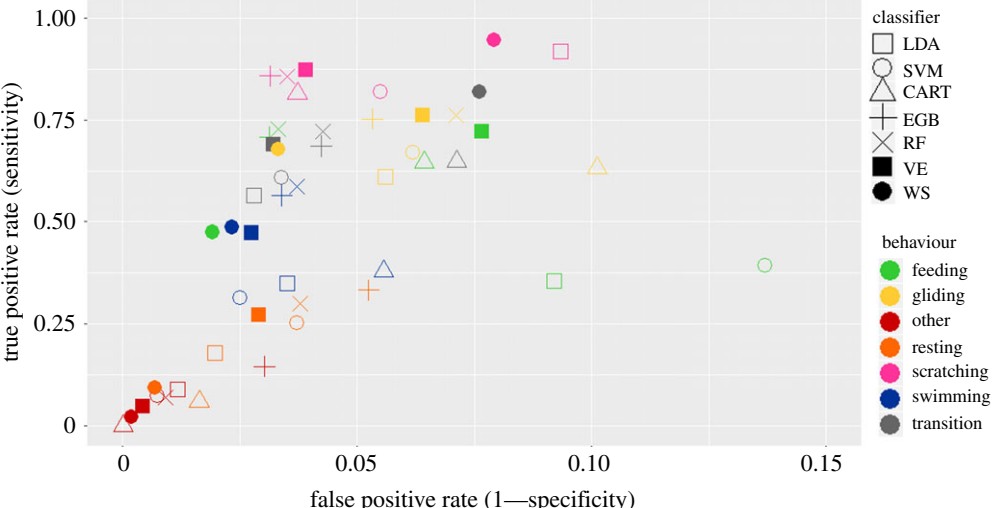

**Figure 4.** True positive rate versus the false positive rate obtained with the seven classifiers for the seven diving categories. The symbols show the mean values obtained from 371 combinations of splitting the sample of 13 individuals into two sub-groups (one of nine individuals for learning and one of four individuals for testing).

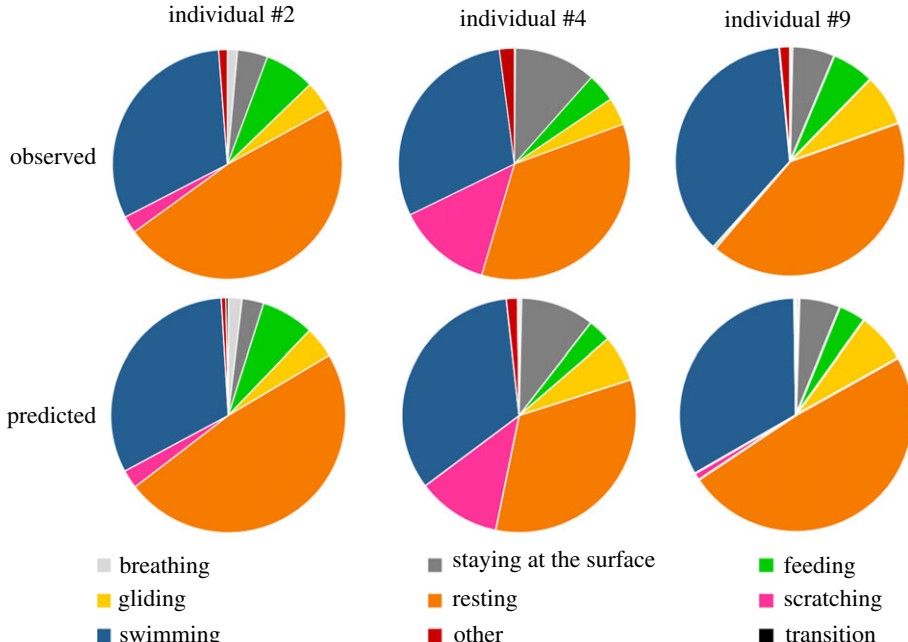

**Figure 5.** Pie chart of the actual (determined from the video) versus predicted mean durations of the various behaviours displayed by three free-ranging immature green turtles. The predicted durations of the diving behaviours were obtained using the WS classifier.

describe swimming behaviours and buoyancy regulation in sea turtles [19,20,47,48] in specific contexts where signals associated with 'Swimming' and 'Gliding' could be visually identified, or were used to estimate sea turtle activity levels in terms of DBA [18,49]. The possibility to rely on accelerometers and other carapace-mounted sensors such as TDRs and gyroscopes to infer behaviours of free-ranging sea turtles had not been explored in detail until now due to the lack of a validation process, which is critically important for this kind of approach [50]. The validation process described in the present study has enabled us to elaborate an overall procedure to reliably infer the seven most commonly expressed behaviours of the free-ranging green turtle (namely 'Breathing', 'Feeding', 'Gliding', 'Resting', 'Scratching', 'Staying at the surface' and 'Swimming'), and thus to infer the fine-scale activity budgets of animals whose populations are currently under anthropogenic pressures which jeopardize their future [51,52]. This inference is essential if we wish to compare how these animals

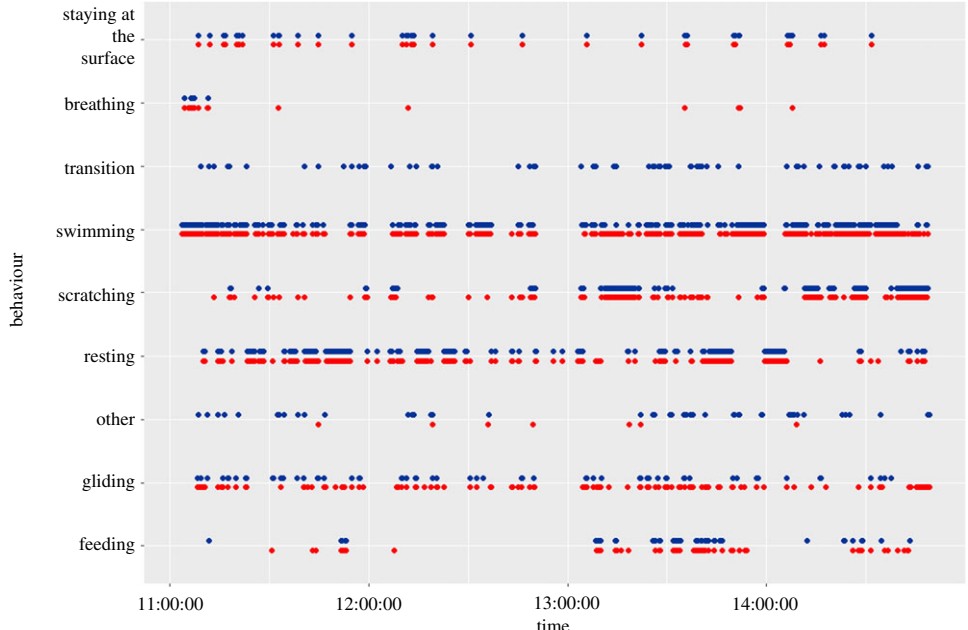

**Figure 6.** Comparison of the nine main inferred behavioural categories (in red) and of the actually observed ones (in blue) for a few hours for immature green turtle #1. The predicted occurrences of the diving behaviours were obtained using the WS classifier.

**Table 2.** Average duration of each behaviour shown by the 13 immature green turtles' predicted time versus actual time. The percentages are expressed with respect to the total individual recorded video duration or to the time the behaviour in question was expressed. The predicted durations of the diving behaviours were obtained using the WS method, and the surfacing behaviours were predicted using depth values.

| behaviour | predicted (s) | observed (s) | difference (s) | %_total | %_behaviour |
|---|---|---|---|---|---|
| breathing | 99 | 94 | 32[a] | 0.2[a] | 46.9 |
| feeding | 432 | 497 | 207 | 1,1 | 180.1 |
| gliding | 936 | 651 | 326 | 1.0 | 39.1 |
| other | 92 | 235 | 143 | 0.8 | 60.2 |
| resting | 9175 | 9640 | 747[b] | 2.7[b] | 6.6 |
| scratching | 437 | 354 | 118 | 0.7 | 311.8[b] |
| staying at the surface | 1435 | 1477 | 151 | 0.8 | 10.8 |
| swimming | 6206 | 6256 | 441 | 2.4 | 6.9[a] |
| transition | 48 | — | 48 | 0.2[a] | — |

[a]The lowest difference obtained among the nine behavioural categories.
[b]The highest difference obtained among the nine behavioural categories.

allocate their time between different activities according to natural and anthropogenic pressures such as available resources, environmental changes or tourism. When combined with GPS data, this protocol may help to identify the areas where sea turtles concentrate their activities and thus help to delineate protected areas in order to limit human disturbances.

We tested seven classifiers (LDA, SVM, CART, RF, EGB, VE and WS) to compare their strengths and weaknesses in automatic behavioural identification based on TDR, acceleration and gyroscopic data. The classifiers identified the seven behavioural classes with a global accuracy ranging from 0.91 to 0.95, which is comparable to the accuracy reached in other similar studies [17,53,54]. The WS classifier performed better than the base and VE classifiers: clearly, assigning precision-based weights to the base classifier predictions improved the behavioural classification. The decrease we observed in the false positive rate for the rare behaviours through the use an ensemble method in this study has also been highlighted by Brewster *et al.* [37]. Ensemble methods are mainly used because they reduce the

variance of behaviour classification [53,55] and thus increase the global accuracy. However, they involve a higher computational cost and require a reliable setting-up of base learners.

The use of supervised machine-learning has become common to automatically identify behaviours from data provided by animal-borne loggers [17,50,56]. Indeed, the development of fast personal computers and of free user-friendly computing libraries made it possible to easily apply these 'black box' algorithms to huge amounts of data. The machine-learning approach has thus turned out to be a very powerful tool for identifying well-characterized behaviours (in terms of signal) such as locomotion [56–58] and resting [59–61]. However, it appears to be rather inefficient when seeking to identify behaviours with confusing signal characteristics. Examples include feeding and grooming in pumas [62], pecking in plovers [63] or foraging in fur seals [64]. Although one could expect that feeding machine-learning algorithms with big data should provide the most accurate predictive rules [16,65,66], Wilson *et al.* [67] showed that a classification method based on a good understanding and careful examination of the acceleration signal actually gives better results in terms of computational time and of accuracy than non-optimized machine learning. Accordingly, the mixed approach developed in this study fed machine-learning algorithms with a number of derived signals which were specifically elaborated to pinpoint specific hard-to-detect behaviours when alternative simpler means based on a single or a few parameters appeared to be effective. This method allowed us to identify key behaviours such as feeding and scratching, which had previously been either misidentified or not identified at all due to the lack of discriminative signals in the raw data obtained from raw acceleration and/or gyroscopic data obtained with loggers fixed to the carapace of the turtle. Although our choice of derived signals makes our approach specific to sea turtles, this principle can be applied to numerous species if the different signals are considered with care before the study.

When carrying out automatic behavioural identification from multi-sensor data using supervised learning algorithms, one of the main difficulties is the segmentation of the multi-sensor data to obtain homogeneous segments that are representative of the various behavioural categories. To date, most studies divided the multi-sensor data into segments using fixed-time segments [68–70] or a sliding sample window with a fixed length [38,71]. However, several studies testing the size of the window showed that it influences classification accuracy and the identification of short behaviours [53,72–74]. Indeed, an individual can express both short and long behaviours, such as burst swimming in lemon sharks or a prey capture in Adélie penguins compared to normal swimming behaviour [37,75]. While the use of long fixed segments dramatically increases the proportion of inhomogeneous segments, using short segments may prevent the detection of certain key signals such as low-frequency oscillations. A hierarchical, adapted segmentation procedure therefore seems to be a more judicious choice. This consists of splitting behaviours into groups based on signals that are easily interpretable in a dichotomic way (variables such as depth were used to attain this in our study). A change-point algorithm can be used to achieve a more specific segmentation based on other signals, with a possible *ad hoc* adjustment of the contrast required to evaluate whether two successive values do or do not belong to the same segment (such as the manual penalty of the PELT algorithm). In this paper, we demonstrate this approach for the green turtle (figure 3, R-script in electronic supplementary material), but there is no reason it could not be easily adapted for other species. This will certainly necessitate the identification of the optimal hyper-parameters as well as the informative signals for the segmentation according to the species, but the approach of combining automated segmentation and machine learning methods with well-thought-out descriptive variables should apply as well.

The approach we proposed thus offers promising perspectives for inferring behaviours of animals that cannot be easily observed in the wild through the automatic analysis of large amounts of raw data acquired over long periods by miniaturized (low-disturbance) loggers such as high-frequency tri-axial accelerometers and gyroscopes. It provides a number of adaptable principles that enable the efficient use of machine learning algorithms to automatically identify fine-scale behaviours in sea turtles, and may be used for a wide range of species. The automated and reliable identification of the various behaviours permits a rapid inference of the time budget of the animals under study. Identifying how much time the studied animals dedicate to activities such as feeding, travelling and resting can be of relevance when seeking to understand how individuals attempt to maximize their fitness in a given environment. This approach should therefore be a key tool in understanding the ecology of endangered species and make a significant contribution to their conservation.

Ethics. This study meets the legal requirements of the countries in which the work was carried out and follows all institutional guidelines. The protocol was approved by the 'Conseil National de la Protection de la Nature' (http://www.avis-biodiversite.developpement-durable.gouv.fr/bienvenue-sur-le-site-du-cnpn-et-du-cnb-a1.html), and the

French Ministry for Ecology, Sustainable Development and Energy (permit no. 2013154-0037), which acts as an ethics committee in Martinique. The fieldwork was carried out in strict accordance with the recommendations of the Prefecture of Martinique in order to minimize the disturbance of animals (authorization no. 201710-0005).

Data accessibility. The R-script to visualize the raw acceleration, gyroscope and depth profile associated with the observed behaviours of the immature green turtles have been uploaded as part of the electronic supplementary material. The same is true for the R-script to automatically identify sea turtle behaviour from the labelled data. The datasets containing the acceleration, gyroscope and depth recordings of the 13 immature green turtles as well as their observed behaviours are available within the Dryad Digital Repository: https://doi.org/10.5061/dryad.hhmgqnkd9 [76].

Authors' contributions. D.C., H.D. and S.R. contributed conception and design of the study. L.J., D.C., J.M., F.S., P.L., J.G., D.E., C.H., Y.L.M., G.H., A.A., S.R., N.L., C.F., C.M., T.M., L.A., G.C., F.L., N.A. and A.B. contributed to data acquisition. L.J., S.G. and S.B. performed the data acceleration analysis and L.J. and V.P.B. the statistical analysis. L.J. wrote the first draft of the manuscript and S.B., V.P.B., F.S. and D.C. contributed critically to subsequent versions.

Competing interests. We declare we have no competing interests.

Funding. This study was co-financed by the FEDER Martinique (European Union, Conventions 2012/DEAL/0010/4-4/ 31882 and 2014/DEAL/0008/4-4/ 32947), DEAL Martinique (Conventions 2012/DEAL/0010/4-4/31882 and 2014/ DEAL/0008/4-4-32947), the ODE Martinique (Convention 014-03-2015), the CNRS (Subvention Mission pour l'Interdisciplinarité), the ERDF fund (Convention CNRS-EDF- juillet2013) and the Fondation de France (Subvention Fondation Ars Cuttoli Paul Appell). Lorène Jeantet's PhD scholarship was supported by DEAL Guyane and CNES Guyane.

Acknowledgements. This study was carried out within the framework of the Plan National d'Action Tortues Marines de Martinique et Guyane Française. The authors also appreciate the support of the ANTIDOT project (Pépinière Interdisciplinaire Guyane, Mission pour l'Interdisciplinarité, CNRS). The authors thank the DEAL Martinique and Guyane, the CNES, the ODE Martinique, the ONCFS Martinique and Guyane, the ONEMA Martinique and Guyane, the SMPE Martinique and Guyane, the ONF Martinique, the PNR Martinique, the Surfrider Foundation, Carbet des Sciences, Plongée-Passion, the Collège Cassien Sainte-Claire and the Collège Petit Manoir for their technical support and field assistance. We are also grateful to the numerous volunteers and free divers for their participation in the field operations. Results obtained in this paper were computed on the vo.grand-est.fr virtual organization of the EGI Infrastructure through IPHC resources. We thank EGI, France Grilles and the IPHC Computing team for providing the technical support, computing and storage facilities. We are also grateful to the three anonymous reviewers for their helpful corrections and comments.

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
