## [Reviewer comments · Royal Society Open Science]

Review History

RSOS-200139.R0 (Original submission)

Review form: Reviewer 1

Is the manuscript scientifically sound in its present form?

Yes

Are the interpretations and conclusions justified by the results?

Yes

Is the language acceptable?

Yes

Do you have any ethical concerns with this paper?

No

Have you any concerns about statistical analyses in this paper?

No

Recommendation?

Accept as is

Comments to the Author(s)

Thanks for your very careful edits of the manuscript. I enjoyed reading it and think you have made a nice contribution. I have only a couple more, extremely minor English edits.

Line 84: replace 'renewed' with 'new', as you are proposing something different than that done previously.

Line 100: replace 'before we can' with 'to'. Simpler.

Line 108: replace 'although' with 'as'. The mean here is that you are doing the video in order to be able to use only the loggers. Alternatively, 'given that' is also fine here.

Review form: Reviewer 2

Is the manuscript scientifically sound in its present form?

Yes

Are the interpretations and conclusions justified by the results?

Yes

Is the language acceptable?

Yes

Do you have any ethical concerns with this paper?

No

Have you any concerns about statistical analyses in this paper?

No

Recommendation?

Accept with minor revision (please list in comments)

Comments to the Author(s)

See attached comments (Appendix A).

Decision letter (RSOS-200139.R0)

31-Mar-2020

Dear Professor Jeantet

On behalf of the Editors, I am pleased to inform you that your Manuscript RSOS-200139 entitled "Behavioural inference from signal processing using animal-borne multi-sensor loggers: a novel solution to extend the knowledge of sea turtle ecology" has been accepted for publication in

Royal Society Open Science subject to minor revision in accordance with the referee suggestions. Please find the referees' comments at the end of this email.

The reviewers and handling editors have recommended publication, but also suggest some minor revisions to your manuscript. Therefore, I invite you to respond to the comments and revise your manuscript.

- Ethics statement

- Data accessibility

If you wish to submit your supporting data or code to Dryad (<http://datadryad.org/>), or modify your current submission to dryad, please use the following link:
<http://datadryad.org/submit?journalID=RSOS&manu=RSOS-200139>

- Competing interests

- Authors' contributions

- Acknowledgements

- Funding statement

Because the schedule for publication is very tight, it is a condition of publication that you submit the revised version of your manuscript before 09-Apr-2020. Please note that the revision deadline will expire at 00.00am on this date. If you do not think you will be able to meet this date please let me know immediately.

Please note that Royal Society Open Science charge article processing charges for all new submissions that are accepted for publication. Charges will also apply to papers transferred to

Royal Society Open Science from other Royal Society Publishing journals, as well as papers submitted as part of our collaboration with the Royal Society of Chemistry (<https://royalsocietypublishing.org/rsos/chemistry>).

If your manuscript is newly submitted and subsequently accepted for publication, you will be asked to pay the article processing charge, unless you request a waiver and this is approved by Royal Society Publishing. You can find out more about the charges at <https://royalsocietypublishing.org/rsos/charges>. Should you have any queries, please contact openscience@royalsociety.org.

on behalf of Dr Asha de Vos (Associate Editor) and Pete Smith (Subject Editor)
openscience@royalsociety.org

Associate Editor Comments to Author (Dr Asha de Vos):

Associate Editor: 1

Comments to the Author:

The reviewers seem happy with the manuscript but request a few revisions that will no doubt make the paper even better. I look forward to seeing the completed product!

Associate Editor: 2

Comments to the Author:

We are happy to send your paper for peer review. Thanks for your submission.

Reviewer comments to Author:

Reviewer: 1

Comments to the Author(s)

Thanks for your very careful edits of the manuscript. I enjoyed reading it and think you have made a nice contribution. I have only a couple more, extremely minor English edits.

Line 84: replace 'renewed' with 'new', as you are proposing something different than that done previously.

Line 100: replace 'before we can' with 'to'. Simpler.

Line 108: replace 'although' with 'as'. The mean here is that you are doing the video in order to be able to use only the loggers. Alternatively, 'given that' is also fine here.

Reviewer: 2

Comments to the Author(s)

See attached comments

Author's Response to Decision Letter for (RSOS-200139.R0)

See Appendix B.

Decision letter (RSOS-200139.R1)

17-Apr-2020

Dear Professor Jeantet,

It is a pleasure to accept your manuscript entitled "Behavioural inference from signal processing using animal-borne multi-sensor loggers: a novel solution to extend the knowledge of sea turtle ecology" in its current form for publication in Royal Society Open Science.

on behalf of Dr Asha de Vos (Associate Editor) and Pete Smith (Subject Editor)
openscience@royalsociety.org

Appendix A

RSOS-200139 Behavioural inference from signal processing using animal-borne multi-sensor loggers: a novel solution to extend the knowledge of sea turtle ecology

General Comments:

This manuscript describes a method to use machine learning to identify specific behaviors of green sea turtles. They use high resolution data to feed the ML algorithms and validate the results using video footage. Overall, the concept of the paper is great, it is well thought out and easy to read. The authors did a good job incorporating comments from previous reviewers and made adjustments accordingly. The paper does provide good information on sea turtle behaviors that were not previously identified; however, I still have a few major concerns before I think the paper is ready for publication.

- 1) A major claim is that the manuscript presents an 'easily adaptable and replicable approach for the long-term automatic identification of different activities and determination of time-budgets in sea turtles'; but the reality of the reproducibility and what long-term implies is not entirely correct. Yes, other researchers can collect accel, gyro, and depth data and use similar machine learning methods, but the behaviors that are identified by their models may be different from what these authors report. Additionally, any high-rate data collection, without the ability to process the data onboard the tag for transmission, will result in short deployment periods (hours to days) so the 'long term' implication that the authors suggest is misleading. If they were able to use the ML results to then identify similar behaviors from instruments at lower sample rates or with just the depth sensor, then there would be a long-term component to discuss. In its current state, this manuscript describes a good methodology for analyzing the fine scale behavior of juvenile green sea turtles, which can then provide more confidence in the activity budgets for those individuals. The authors should tone down the bold claims and simply describe the method and results without attempting to make this bigger than it really is. These methods have been used in other species for similar purposes so the novelty of the method itself isn't there, what makes this unique is that they are incorporating the gyro data, which other researchers decided not to use since accel data on its own was sufficient for their studies.
- 2) The take-home message gets lost. They go through extensive data processing and modeling with the goal of (I think) to better define activity budgets for juvenile green turtles, but the last step in the story (the actual activity budgets and the implications thereof) is very weak and does not feel fully developed. Either refocus the message in the introduction to be on the method itself, or spend more time developing the implications of the better activity budgets that are produced using this method.
- 3) The benefits of including the gyro data, which is what makes this method unique, are not discussed. Did the data help with the classification? Did it allow for detection of different behaviors? Should other people be using the gyro data even though it's expensive (battery-wise)? The paper does not need to be unique for publication, but since that is one of the claims the authors are making, I think it warrants some discussion on why the gyro data is helpful and worth including.

Specific Comments:

Line 69: This is one of the main claims of the introduction. 'proportion of time allocated to feeding on the long term remains unknown'. What do you mean by long term? Any high-rate sampled data will still

only be hours to days in duration. Either define what you mean, or adjust the message to something more appropriate for this type of data collection.

Line 130: Change to: ...combined with a tri-axial accelerometer and gyroscope, and a TDR...

Line 133-138: A few things to note here. First, reporting the bits is more of an engineering unit to know how many decimal places are used and is unnecessary information for a manuscript. Second, the CATS website shows different sensor specifications than what the authors report. If they used a non-standard depth sensor with a 0-100m depth range instead of the standard 0-2000m range then they should report that. Currently, line 138 says that depth was recorded at 1Hz in the range of +/- 100m. That just doesn't make sense in this context since the pressure sensor will report from 0-100, not -100 to +100. If I'm correct in my interpretation and that the authors did get a special pressure sensor, then I would reword this section something like: These devices were programmed to record acceleration and rotational velocity (gyroscope) at 20 or 50 Hz. Depth was recorded at 1Hz using a pressure sensor with a range from 0-100m and 0.2m accuracy.

Line 147: geolocation is misspelled.

Line 149-151: Reword this. One suggestion would be: Instruments were deployed on 37 individuals, but complete datasets including video, acceleration, gyroscope, and depth values were only recovered for 13.

Section 2.4: Dynamic Body Acceleration is a very common calculation so the details here are unnecessary. You could simplify this quite a bit by saying you used a 2 sec window to calculate the DBA. Then just show the bold equations in order.

Line 185: I really like that you split the data into behavior bouts.

Line 187: Using 0.3m as the start of a dive definition may be a little suspect. With 0.2m accuracy in the pressure sensor, a threshold value of 0.3m may be too shallow to be accurate. However, this would depend on how CATS calculates the accuracy. If it refers to a standard deviation in the recorded values then it may be okay, but if it refers to a bound on the total error, then 0.3m is too shallow of a value to use as a threshold since it would be hard to know if that animal/tag was actually at that level or not. I would check in with CATS to verify what they mean by accuracy with regard to their pressure sensor and then proceed accordingly.

Line 209: I'm just curious how you came up with the 3/5 qualifier?

Line 226: using the accel, gyro, DBA, and RA values creates a lot of correlation between your variables. Can you comment on how this may have impacted the results? Did you try it with just DBA and RA? 42 is a lot of variables to include. Did you do any sort of selection to see if all of them were necessary?

Line 265: remove the words 'so-called'.

Line 294: I think the word 'discriminated' is not what you mean to use here? Are you trying to say that the seven classifiers identified the five behavioral categories?

Line 305: I think 'Activity Budget' is a more common phrase for what the authors are describing here. Also, it would be nice to have a sentence or two describing what the general activity budgets were. (e.g.

the majority of time was spent Swimming and Resting, with Feeding and Scratching occurring least often for all individuals...or something similar).

Line 307: Change to: This result highlights the ability of our method and the WS model to predict the behavior of immature green sea turtles in natural conditions.

Line 308: Change to...The main differences between the observed and predicted activity budgets were seen in the Resting and Swimming behaviors. These differences were small and represented less than 3% of the total observed time. Feeding and Scratching were under-represented in our models and consequently represent roughly 1% of the total observed time.

Line 345: Delete 'Unsurprisingly...' Change to: The WS classifier performed better than...

Appendix B

Editors :

Associate Editor: 1

Comments to the Author:

The reviewers seem happy with the manuscript but request a few revisions that will no doubt make the paper even better. I look forward to seeing the completed product!

Associate Editor: 2

Comments to the Author:

We are happy to send your paper for peer review. Thanks for your submission.

We thank the editors for considering our manuscript and accepting the revised version with great enthusiasm. We are convinced that this article will be useful to the scientific community working on automatic behavioural identification.

Reviewer: 1

Thanks for your very careful edits of the manuscript. I enjoyed reading it and think you have made a nice contribution. I have only a couple more, extremely minor English edits.

Line 84: replace 'renewed' with 'new', as you are proposing something different than that done previously.

Line 100: replace 'before we can' with 'to'. Simpler.

Line 108: replace 'although' with 'as'. The mean here is that you are doing the video in order to be able to use only the loggers. Alternatively, 'given that' is also fine here.

We thank the referee for reviewing our manuscript and we appreciate that he/she enjoyed reading it. We took into account his/her advice for the language issues and corrected the sentences.

Reviewer: 2

General Comments:

This manuscript describes a method to use machine learning to identify specific behaviors of green sea turtles. They use high resolution data to feed the ML algorithms and validate the results using video footage. Overall, the concept of the paper is great, it is well thought out and easy to read. The authors did a good job incorporating comments from previous reviewers and made adjustments accordingly. The paper does provide good information on sea turtle behaviors that were not previously identified; however, I still have a few major concerns before I think the paper is ready for publication.

We thank the referee for carefully reading our manuscript and pointing out significant improvements. Here our answer about his/her main concerns:

1) A major claim is that the manuscript presents an 'easily adaptable and replicable approach for the long-term automatic identification of different activities and determination of time-budgets in sea turtles'; but the reality of the reproducibility and what long-term implies is not entirely correct. Yes, other researchers can collect accel, gyro, and depth data and use similar machine learning methods, but the behaviors that are identified by their models may be different from what these authors report. Additionally, any high-rate data collection, without the ability to process the data onboard the tag for transmission, will result in short deployment periods (hours to days) so the 'long term' implication that the authors suggest is misleading. If they were able to use the ML results to then identify similar behaviors from instruments at lower sample rates or with just the depth sensor, then there would be a long-term component to discuss. In its current state, this manuscript describes a good methodology for analyzing the fine scale behavior of juvenile green sea turtles, which can then provide more confidence in the activity budgets for those individuals. The authors should tone down the bold claims and simply describe the method and results without attempting to make this bigger than it really is. These methods have been used in other species for similar purposes so the novelty of the method itself isn't there, what makes this unique is that they are incorporating the gyro data, which other researchers decided not to use since accel data on its own was sufficient for their studies.

We do think that the novelty of the manuscript comes from the approach itself (rather than the use of the gyroscope) which combines different sensors, careful inspection of the data, derived signals specifically elaborated and ML. As the referee mentioned it, the ML has been used in other species but most of the time without upstream identification of the informative characteristics of the data streams. As a result, the algorithms are very powerful to identify well discriminated behaviours but less efficient for those with confusing signal characteristics. In the paper, we therefore attempted to propose the best of the two worlds: at the first stage, we used obvious signals to sort surfacing and diving behaviours. At the second stage, we relied on an automated "craft" approach to deal with the signals so as to isolated homogeneous behavioural bouts. At the third stage, we used a machine learning approach to characterise the different types of behaviours by a unique combinations of the signal values. By replicable, we mean that the principle of this approach can be applied to many species, but do not mean that the exact same method could be applied directly without adaptations to the particular characteristics of these other species. To be crystal clear on this point, we added a sentence about this at the end of the discussion section (L. 400-404).

About the term "long term", there is no particular technical limitation for recording acceleration, depth, or angular velocity at high frequency for several weeks, even if "long-term" deployments in sea turtles are limited by the difficulty of recovering the logger. The problem with the term "long-term" is probably that it is interpreted differently by different people. To clarify this in the present context, we specified that by "short-term" vs. "long-term" we mean "a few hours" (L. 87) vs. "up to a few weeks" (L 93) in the revised version.

2) The take-home message gets lost. They go through extensive data processing and modeling with the goal of (I think) to better define activity budgets for juvenile green turtles, but the last step in the story (the actual activity budgets and the implications thereof) is very weak and does not feel fully developed. Either refocus the message in the introduction to be on the method itself, or spend more time developing the implications of the better activity budgets that are produced using this method.

We agree that there is some discrepancy between the introduction and the discussion sections. Clearly, this paper aims at providing a better methodological approach rather than new insights about the biology of sea turtles. Accordingly, in the revised version of the paper, we specified this aim more clearly in the introduction section (l. 99-106).

3) The benefits of including the gyro data, which is what makes this method unique, are not discussed. Did the data help with the classification? Did it allow for detection of different behaviors? Should other people be using the gyro data even though it's expensive (battery wise)? The paper does not need to be unique for publication, but since that is one of the claims the authors are making, I think it warrants some discussion on why the gyro data is helpful and worth including.

As mentioned above, even if the gyroscope is scarcely used in behavioural identification, we do not think that this is its use that makes our approach particularly interesting. Accordingly, we preferred insisting on the other aspects such as the combination of various sensors and methods in a consistent, unified methodology. For some species, such as sea turtles, the use of a gyroscope was necessary in our study because we were not able to properly detect some behaviours without it, but it will not be necessarily the case for other species. We understood the previous version of our paper may have been misleading about the relevance of the use of the gyroscope in sea turtles, so we rephrased the last paragraph of the introduction to avoid putting gyroscope use on the front (L106-109).

Specific Comments:

-Line 69: This is one of the main claims of the introduction. 'proportion of time allocated to feeding on the long term remains unknown'. What do you mean by long term? Any high-rate sampled data will still only be hours to days in duration. Either define what you mean, or adjust the message to something more appropriate for this type of data collection.

As mentioned above, we clarified this term (as well as short-term) in the context of our study (L. 87 and 93).

-Line 130: Change to: ...combined with a tri-axial accelerometer and gyroscope, and a TDR...

Done

-Line 133-138: A few things to note here. First, reporting the bits is more of an engineering unit to know how many decimal places are used and is unnecessary information for a manuscript. Second, the CATS website shows different sensor specifications than what the authors report. If they used a non-standard depth sensor with a 0-100m depth range instead of the standard 0-2000m range then they should report that. Currently, line 138 says that depth was recorded at 1Hz in the range of +/- 100m. That just doesn't make sense in this context since the pressure sensor will report from 0-100, not -100 to +100. If I'm correct in my interpretation and that the authors did get a special pressure sensor, then I would reword this section something like: These devices were programmed to record acceleration and rotational velocity (gyroscope) at 20 or 50 Hz. Depth was recorded at 1Hz using a pressure sensor with a range from 0-100m and 0.2m accuracy.

We thank the reviewer for pointing this and corrected the text accordingly (L. 137-141)

-Line 147: geolocation is misspelled.

Right. Corrected.

-Line 149-151: Reword this. One suggestion would be: Instruments were deployed on 37 individuals, but complete datasets including video, acceleration, gyroscope, and depth values were only recovered for 13.

We reworded the sentence as suggested (L. 152-154).

-Section 2.4: Dynamic Body Acceleration is a very common calculation so the details here are unnecessary. You could simplify this quite a bit by saying you used a 2 sec window to calculate the DBA. Then just show the bold equations in order.

This section was dramatically shortened (L. 172-179)

-Line 185: I really like that you split the data into behavior bouts

Thanks.

-Line 187: Using 0.3m as the start of a dive definition may be a little suspect. With 0.2m accuracy in the pressure sensor, a threshold value of 0.3m may be too shallow to be accurate. However, this would depend on how CATS calculates the accuracy. If it refers to a standard deviation in the recorded values then it may be okay, but if it refers to a bound on the total error, then 0.3m is too shallow of a value to use as a threshold since it would be hard to know if that animal/tag was actually at that level or not. I would check in with CATS to verify what they mean by accuracy with regard to their pressure sensor and then proceed accordingly.

We do not know the details about the CATS TDR. We simply set this threshold empirically, as it made it possible to well distinguish between the surface and diving behaviours (in agreement with the video footage)

-Line 209: I'm just curious how you came up with the 3/5 qualifier?

This ratio was also established empirically. It corresponded to the best trade-off between the number of "Transition" segments (which bring confusion to ML) and the number of segments assigned with a behavioural category. As the "Feeding" behaviour was rarely expressed and difficult to segment from the resting sequences, we found that a lower and less strict threshold in obtaining the Transition segments led to a better discrimination of "Feeding" by the ML.

-Line 226: using the accel, gyro, DBA, and RA values creates a lot of correlation between your variables. Can you comment on how this may have impacted the results? Did you try it with just DBA and RA? 42 is a lot of variables to include. Did you do any sort of selection to see if all of them were necessary?

There may indeed exist a lot of correlations between some of the variables. This is often a source of problems for some statistical techniques such as GLM. However, for Machine Learning the situation is quite different: the model selection procedure is the algorithm training, which sets the importance of the input variables, and the only serious problem of highly correlated variables could be that different combinations of model hyper-parameters lead to the same classification result. Since we are interested in the classification itself and not in the study of the hyper-parameters, the correlation is not an issue at all. This is why 42 features is not that large (other studies do not hesitate in adding as many variables as available, sometimes involving several thousand features). Moreover, because we used a cross validation approach, we also minimized the risk of overfitting. We tested several combinations of features (or Principal Components) to find a reduced set that results in the same accuracy with a reduced number of variables, but we could not find one. The reason is probably that we classified many different behaviours, and even if for each of them we could find a reduced set, no such reduced system works for all behaviours at once. This is why we kept all 42 variables. This explanation is summarized L. 237-243 in the revised version.

-Line 265: remove the words 'so-called'.

Done

-Line 294: I think the word 'discriminated' is not what you mean to use here? Are you trying to say that the seven classifiers identified the five behavioral categories?

Indeed, "identify" is more appropriate in this sentence.

-Line 305: I think 'Activity Budget' is a more common phrase for what the authors are describing here. Also, it would be nice to have a sentence or two describing what the general activity budgets were. (e.g. the majority of time was spent Swimming and Resting, with Feeding and Scratching occurring least often for all individuals...or something similar).

We agree and replaced "Time budget" by "Activity budget" in the revised version. These budgets are however not representative of the actual distribution of the activities in the present study because of possible disturbances linked to the recent capture of the animal and the presence of a large device (the video recorder) and which may cause the low frequency of "Feeding" occurrences. The aim of this study was purely methodological. Behavioural studies resting on this approach can be developed in a second stage, involving long-term deployments of small devices (so avoiding the possible stress effects recent capture and large device). This is why, as already mentioned above, we chose not to describe and interpret the activity budgets obtained in the present study.

-Line 307: Change to: This result highlights the ability of our method and the WS model to predict the behavior of immature green sea turtles in natural conditions.

Done

-Line 308: Change to...The main differences between the observed and predicted activity budgets were seen in the Resting and Swimming behaviors. These differences were small and represented less than 3% of the total observed time. Feeding and Scratching were under-represented in our models and consequently represent roughly 1% of the total observed time.

Done

-Line 345: Delete 'Unsurprisingly...' Change to: The WS classifier performed better than...

Done